# Influence of Environmental Factors on Prey Discrimination of Bait-Attracted White Sharks from Gansbaai, South Africa

**DOI:** 10.3390/ani12233276

**Published:** 2022-11-24

**Authors:** Francesca Romana Reinero, Emilio Sperone, Gianni Giglio, Antonio Pacifico, Makenna Mahrer, Primo Micarelli

**Affiliations:** 1Sharks Studies Center-Scientific Institute, 58024 Massa Marittima, Italy; 2Department of Biology, Ecology and Earth Sciences, University of Calabria, 87036 Rende, Italy; 3Department of Political Science and CEFOP-LUISS, LUISS Guido Carli University, 00197 Rome, Italy; 4W. M. Keck Science Department, Claremont McKenna College, Claremont, CA 91711, USA; 5Department of Physical Sciences, Earth and Environment, University of Siena, 53100 Siena, Italy

**Keywords:** elasmobranch, environmental effects, predatory behavior, prey choice, sensory ecology

## Abstract

**Simple Summary:**

Predator–prey interactions can be influenced by environmental factors and depend on the sensory capabilities of a predator and its prey. Environmental factors influence the prey discrimination of white sharks in Gansbaai during their passive prey predatory behavior. Tide range is the most important factor that influences the white sharks’ prey choice, followed by underwater visibility, water temperature, and sea conditions. With high tide, better underwater visibility, cooler water temperature, and better sea conditions, sharks choose the energetical richer prey, the seal-shaped decoy, instead of the tuna bait. This study confirms the importance of visual ability in mature and immature white sharks with dietary shifts in their feeding pattern.

**Abstract:**

The influence of environmental factors on prey discrimination of bait-attracted white sharks was studied over a six-year period (2008–2013) at Dyer Island Nature Reserve (Gansbaai, South Africa). Across 240 bait-attracted feeding events observed in this period, both immature and mature white sharks were attracted by the seal-shaped decoy rather than the tuna bait, except for the years 2008 and 2011. Tide ranges, underwater visibility, water temperature, and sea conditions were, in decreasing order, the factors which drove white sharks to select the seal-shaped decoy. High tide lowered the minimum depth from which sharks could approach seals close to the shore, while extended visibility helped the sharks in making predatory choices towards the more energy-rich prey source, the odorless seal-shaped decoy. On the contrary, warmer water is associated with an increase in phytoplankton that reduces underwater visibility and increases the diversity of teleosts including tuna—a known prey of white sharks—driving the sharks to favor the tuna bait. Overall, sea conditions were almost always slightly rough, ensuring a good average underwater visibility. Recommendations for future research work at this site are presented.

## 1. Introduction

It is relatively simple to study the white shark (WS) *Carcharodon carcharias* (Linnaeus, 1758) predatory behavior, thanks to the ease with which it can be observed from the surface attacking and feeding on pinniped colonies close to rocky islands, where white sharks congregate [1]. Predator–prey interactions can be influenced by environmental factors and depend on the sensory capabilities of a predator and its prey [2]. Accordingly, these factors can drive the former’s ability to detect the prey, the latter’s ability to avoid the attack, behavioral patterns, and predator activity peaks within marine ecosystems [3]. Studying environmental factors that can affect predator–prey interactions and predation frequencies is also important, considering the populations of many predators are declining globally [4]. Although several works describe the surface predatory behavior [5,6,7,8,9] and the passive prey discrimination of WSs when bait-attracted [10,11,12], studies pertaining to the influence of environmental factors on predator–prey interactions are rare [2,13] and have never specifically focused on bait attraction.

Abiotic factors, such as tide ranges, light levels, underwater visibility, water temperature, and sea conditions, are thought to influence the sharks’ predatory behavior directly through their effects on physiology and sensory ecology, limiting or improving their access to prey [14].

Vision plays an important role in feeding patterns [10,15]. Under normal conditions, seals’ visual acuity can detect WSs from the surface. However, light levels could help sharks to hide from their prey when hunting from the bottom, giving them the possibility of ambushing from the depths to the surface under low light conditions [5,12,16]. Furthermore, Huveeners et al. [17] stated that, at Neptune Island (Australia), sharks hunt near the surface swimming with the sun directly behind them, which hides them in shadow while illuminating the prey ahead. Follows et al. [13], at Seal Island (South Africa), stated that out of 1.476 recorded natural predation events, the white shark attack frequencies and seal capture success rates were higher during periods of high lunar illumination, when seal silhouettes along the surface were easier to discern.

At the same time, olfactory bulbs of WSs comprise 18% of their total brain mass, suggesting the importance of this sense for feeding activity [18]. Winds, which have a direct effect on sea conditions, can propel chemical stimuli, such as blood, excreta, or other organic fractions, that can enable sharks to locate their prey [2]. In addition, northerly winds that predominate during the South African winter storms force seals to return to Seal Island, swimming against the waves and current and producing sounds that aid in prey detection [2].

Tidal height can also influence predation success, since high tide reduces both the minimum depth from which sharks can approach seals and their haul-out area by forcing pinnipeds to go into the water, increasing white shark prey availability [2].

Water temperature, on the other hand, is considered one of the most important abiotic environmental factors that influences the distribution, movements, foraging, and reproductive strategies of sharks, playing a significant role in determining seasonal prey distribution and abundance [14]. Warmer water, which can result in blooms of diatoms, for example, is associated with an increase in abundance and diversity of teleosts and chondrichthyans that are in turn WSs prey [14]. Since it is widely known that WSs are endothermic and able to regulate their internal body temperature, maintaining metabolic rates though specialized circulatory mechanisms [19], the abiotic factor of water temperature for this species has a greater influence on prey distribution and abundance rather than on a shark’s physiological water preference.

The present paper looks more deeply into the previous work of Micarelli et al. [10] at Gansbaai (South Africa), where the authors investigated the surface predatory behavior and the passive prey discrimination of bait-attracted WSs between 2008 and 2013. They observed 240 predator–prey interactions and highlighted that both immature and mature WSs were attracted to the seal-shaped decoy rather than the tuna bait, except during the years 2008 and 2011. Authors ran a non-parametric statistical test, Cochran’s Q test, displaying that the sharks’ predatory choice was not random (rejecting the null) and citing a difference in the effectiveness of sharks’ prey preferences. This finding was likely due to the dietary shift that occurs in WSs 200–340 cm in total length (TL), as well as the sharks’ ability to visually locate and select the highest energy prey source. The latter point was also confirmed in immature sharks.

The aim of this study was to assess if, between 2008 and 2013, environmental factors, such as tide ranges, light levels, underwater visibility, water temperature, and sea conditions, could have influenced the choice of passive prey (tuna bait and seal-shaped decoy) by bait-attracted WSs from Gansbaai, and, in particular: (i) prove that the sharks’ prey preferences are not random but affected by potential not directly observed and measured environmental factors varying over the time; (ii) highlight the main environmental factors causing the WSs to prefer the seal-shaped decoy instead of the tuna bait; and (iii) evaluate major environmental factors affecting predatory behavior. These aspects aim to confirm the influence of environmental factors on the feeding behavior of WSs.

## 2. Materials and Methods

In Gansbaai (South Africa), at the Dyer Island Nature Reserve, a large WSs population with a prevalence of immature individuals is present and can be observed thanks to the support of local ecotourism operators authorized to reach the field observation sites [1]. Dyer Island Nature Reserve is located 7.5 km south-east of Gansbaai (34°41′ S; 19°24′ E) and includes two islands: Dyer Island, a low-profile island ca. 1.5 km long and 0.5 km wide, and Geyser Rock, a small islet ca. 0.5 km long and 180 m wide, characterized by different seabird colonies and a colony of Cape fur seals *Arctocephalus pusillus pusillus* (Schreber, 1775), respectively, (Figure 1).

The methodology follows and goes further than the one used by Micarelli et al. [10]. The study period, from 2008 to 2013 in the autumn season between March and May, required a total of 247 h of effort (on average 41 h/y). Environmental observations and data collection were performed from the 13 m “Barracuda” boat owned by the “Unlimited Shark diving”, anchored at 100–150 m off Dyer Island. An anti-shark cage was fixed to the side of the boat for the entire duration of the observations. To attract sharks to the boats, olfactory stimulants (chum) composed of sea water, cod liver oil (*Gadus* sp.), tuna blood, and small pieces of fish were used, according to the methods described by Laroche et al. [20], Ferreira and Ferreira [21], Strong et al. [22], and Sperone et al. [23].

Passive target preys chosen to attract sharks for passive prey discrimination and predatory observations were an odorless seal-shaped decoy and odorous buoy floating baits of tuna pieces (tuna bait), both of similar size (70 cm long and 32 cm wide juvenile Cape fur seal decoy and 60 cm in diameter floating tuna baits), in line with the research protocol adopted by Sperone et al. [10,15]. Both passive target preys were positioned at the bow 10 m from each other with the aim of testing that the odorless seal-shaped decoy remained isolated from the odorous tuna bait.

Sharks were individually identified, and environmental data were collected by the same Sharks Studies Center—Scientific Institute (Massa Marittima, Italy) research team for the duration of the study. Sometimes a specimen was observed several times throughout the day and all exhibited behaviors were recorded. Sharks’ TL and sex were estimated according to the known size of the cage length and the pelvic fin area observations (males if claspers were seen; females if the lack of claspers was verified and their pelvic fin area was filmed from the cage), respectively [1]. All other specimens were categorized as unknown sex. White sharks’ size at sexual maturity was estimated, according to Hewitt et al. [24] and Micarelli et al. [1]: TL ≥ 450 cm for mature females and TL ≥ 350 cm for mature males. Sex was estimated by a maximum of three observers at a time from the cage, while both passive prey discrimination of bait-attracted WSs and TL were estimated by the same operator from the lower or upper deck of the boat.

During the observations, the following environmental factors were recorded by the same operator every three hours from arrival at the sampling area until the end of daily observations:
(1)Sea conditions were divided according to the Douglas scale of wave height [25]:
(a)“Calm”, which included glassy and rippled (0–10 cm wave height).(b)“Slightly rough”, which included smooth, slight, and moderate (11–250 cm wave height).(c)“Rough”, which included rough, very rough, high, very high, and phenomenal (>250 cm wave height).(2)Light levels were expressed in oktas, a unit of measurement that indicates the cloudiness of the sky, estimated in terms of how many eighths of it are obscured by clouds [26]. Measurement intervals used to assess the sky coverage were as follows:
(a)0–2 oktas corresponded to clear sky.(b)3–5 oktas corresponded to partly cloudy sky.(c)6–8 oktas corresponded to a totally covered sky.(3)Tide ranges were divided into (a) low and (b) high classifications, obtained from the Windguru Database (www.windguru.cz, accessed on 1 June 2015) for the “Kleinbaai, Marine Dynamics” area.(4)Water temperature, expressed in degrees Celsius (°C), was obtained by the boat’s instruments.(5)Underwater visibility, expressed in meters (m), was calculated by operator with the Secchi disc.

The empirical methodology aims to prove that: (i) both passive preys (seal-shaped decoy and tuna bait) do not show spatial autocorrelation; (ii) and the sharks’ passive prey discrimination is affected by potential environmental factors. (i) To measure the overall spatial autocorrelation between passive preys, Moran’s test has been assessed. In our study, the spatial autocorrelation is multi-dimensional—accounting for multiple data objects in the analysis—and then we need to test if the observations are not independent. More precisely, according to Moran’s test, we are able to measure how one object is similar to the others surrounding it. The possible results are equal to: −1, indicating perfect clustering of dissimilar values (perfect dispersion); 0, denoting no autocorrelation (perfect randomness); and +1, indicating perfect clustering of similar values (no perfect dispersion). Thus, we construct a hypothesis testing where the null is in the data that are randomly distributed, while the alternative is that the data are more spatially clustered. Calculations for Moran’s test are based on a weighted matrix, with units *i* and *j*, where i ≠j. Similarities between units are computed as the product of the differences between *y*_i_ and *y*_j_, with *y*_l_ denoting the variable of interest and l=i,j. In this context, *y*_l_ denotes the tuna bait and the seal-shaped decoy for all different units. The formula to compute the test statistic is:$$z_{test}=\frac{1}{s^{2}}\times \frac{\sum_{i}\sum_{j}\left(y_{i}-\bar{y}\right)\left(y_{j}-\bar{y}\right)}{\sum_{i}\sum_{j}w_{ij}}$$
where $s^{2}=\frac{1}{n}\sum_{i}{\left(y_{i}-\bar{y}\right)}^{2}$ denotes the sample variance, *w_ij_* the weighted matrix, and *z*_test_ refers to the test statistic approximatively distributed as a normal standard with a sufficiently large sample (n≥30). (ii) In this study, five main factors assessed during 240 interactions, spanning the period 2008–2013 are considered. They consist of two not directly observed variables and three not directly measured factors split, respectively, into: (i) light levels, measured in oktas; (ii) water temperature, measured in degrees Celsius (°C); (iii) underwater visibility, measured in meters (m); (iv) sea conditions, classified as calm, slightly rough, and rough; and (v) tide, classified as high or low.

The last two elements have been evaluated through proxy discrete variables. More precisely, an ordinal variable was used to assess sea conditions, assuming values 1 (whether the sea is calm), 2 (whether the sea is slightly rough), and 3 (whether the sea is rough). Tide was computed through a dummy variable equal to 1 (low tide) or 0 (high tide).

The water temperature has been grouped in three classes to analytically evaluate its effects on the prey preference and potential interactions with the other covariates. The related discrete variable was then equal to 1 for the class 11.0–13.9 °C; 2 for the class 14.0–16.9 °C; and 3 for the class 17.0–19.9 °C.

The variable of interest denotes sharks’ prey preferences and was measured by means of a dummy variable of either 0, where the sharks preferred the bait, or 1, where the choice corresponded to the seal-shaped decoy.

The statistical methodology consists of a three-step approach called the Three-Step System Multivariate Classification (TSMC). This method combines a first-step non-parametric statistical test for matching heterogeneous groups of variables with a supervised machine learning (ML) technique for consistently estimating all parameters of interest in a multivariate non-linear context.

The first step evaluates a permutational multivariate analysis of variance (PERMANOVA) that can: (i) evaluate potential heterogeneity among units; (ii) investigate heterogeneous effects among factors and outcomes (prey preferences); and (iii) deals with potential endogeneity issues. Let $X_{ik}$ be a N∗K matrix, with i=1,…,N and k=1,…,K denoting units and factors, respectively. We estimate a PERMANOVA between the elements ‘years’ (t=1,…,T) and ‘effects’ (*k*) across each sample-observed unit. The null hypothesis stands for homogeneity among groups (the dispersion is equivalent for all groups), while a rejection of the null highlights that the spread of any two variables investigated in the experiment is different between the groups. PERMANOVA was estimated between the years and environmental effects across each sample-observed unit. The variables are labelled as so: (i) ‘oktas’ denoting light levels; (ii) ‘sea’ referring to sea conditions; (iii) ‘tide’ referring to tide ranges; (iv) ‘visibility’ denoting underwater visibility; and (v) ‘temp.’ describing water temperature grouped across classes. The variable of interest is defined as ‘prey’ and accounts for sharks’ prey preferences (tuna bait or seal-shaped decoy). The main assumptions held are: (i) variables in the dataset are exchangeable under the null; (ii) exchangeable variables, such as sites, observations, and factors, are independent; and (iii) exchangeable variables have similar multivariate dispersion (each unit of observation has a similar degree of multivariate scatter).

Given that every environmental factor observed over time significantly affects the outcome, the second step estimates a logit regression analysis to predict the categorical dependent variable (prey preferences) through a set of covariates. It is one of the most popular supervised ML algorithms because it can provide probabilities and classify non-homogeneous data using continuous and discrete datasets (as in this study). Let $P_{i}$ be the probability that $y_{i}=1$ (preference about seal shape), conditional to the information set $\Omega_{i}$, which contains exogenous and quantitative variables, a model for binary response that serves to model this conditional probability. Notably, $P_{i}$ corresponds to the binary response model thought of as modelling the conditional expectation and can be expressed as:(1)$$P_{i}\;\equiv \mathrm{Pr}\left(y_{i}=1|\Omega_{i}\right)=E(y_{i}|\Omega_{i})$$

In that context, we cannot use the linear model because a linear regression on X fails to impose the condition $0\leq E(y_{i}|\Omega_{i})\leq 1$. To solve (1), we need to impose the constraint by a proper functional form equal to:(2)$$P_{i}\;\equiv \mathrm{Pr}\left(y_{i}|\Omega_{i}\right)=F\left(X_{i}\beta \right)$$
where $X_{i}$ is the matrix defined in the first step—stacked for k—containing all the covariates, the K∗1 vector $\beta_{k}$ is an unknown parameter estimated to predict $y_{i}$, and F() is a logistic function with the form:(3)$$P_{i}\;\equiv \frac{e^{X_{i}\beta }}{1+e^{X_{i}\beta }}$$

The logit model can be written as:(4)$$log\left(\frac{P_{i}}{1-P_{i}}\right)=X_{i}\beta$$
which means that the logarithm of the odd ratio is equal to $X_{i}\beta$.

The last step consists of computing the sample’s marginal effects. The main aim is to assess, in order of importance, the major predictors affecting the prey preferences across units over time. Let the logit model be expressed by Equation (4), changes in the values of $X_{i}$ affecting $E(y_{i}|\Omega_{i})$ in a non-linear fashion can be computed by:(5)$$\frac{\partial P_{i}}{\partial x_{ik}}=\frac{\partial F\left(X_{i}\beta \right)}{\partial x_{ik}}=f\left(X_{i}\beta \right)*\beta_{k}\;$$

For most transformation functions used, $f\left(X_{i}\beta \right)$ reaches its maximum at $X_{i}\beta =0$, meaning that the effect of a change in $x_{ik}^{\prime }$ on $P_{i}$ is at maximum when $P_{i}=0.5$ and reduced when $P_{i}$ is close to 0 or 1. Given (potential) unobserved interactions among factors, a White’s correction for the presence of heteroskedasticity in the calculation of marginal effects standard errors is applied, obtaining larger significance and lower AIC (Akaike information criterion (AIC) is usually used to match similar regression functions and choose the best one).

## 3. Results

### 3.1. Moran’s Test

The Moran’s test statistic obtained was equal to −3.93 and, in absolute value with two-side alternative hypothesis, was bigger than the critical value $z_{\frac{\alpha }{2}}=z_{0.025}=1.96$, with α=5% as default. Thus, the hypothesis testing is significant and, letting the statistic be negative, denotes perfect dispersion between the tuna bait and the seal-shaped decoy.

### 3.2. PERMANOVA Test

The first step, the PERMANOVA Test (Table 1), proves that the sharks’ prey preferences are not random but are affected by potential that is not directly observed and measured environmental factors varying over time.

Table 1 displays the estimation outputs, where the significance level is α=5% (as default). Almost all predictors are significant (except ‘visibility’), displaying a *p*-value close to zero and lower than α, and the residuals are null, highlighting the robustness and validity of the results. Looking closely at the estimates, two findings should be dealt with carefully: (i) according to the predictor ‘visibility’, it would not be significant across units over time even if its related sum of squared dissimilarities and Pseudo-F statistic were null and highly large, respectively; (ii) the predictor ‘oktas’ displays, on average, larger *p*-values and lower Pseudo-F than the other significant ones. These findings highlight that multicollinearity problems would matter. For instance, the factor ‘visibility’ could be strongly correlated with one or more predictors within the model, rejecting the assumption in (ii) (exchangeable variables, such as sites, observations, and factors, are independent). To deal with this, we ran a new PERMANOVA test dropping ‘oktas’, the factor displaying lower Pseudo F statistic, and obtained the expected result (Table 2). The variable ‘visibility’ becomes significant with lower SS and higher Pseudo-F, compared to before. According to the other predictors, the estimated results improve as well. Thus, the estimates are now valid and unbiased.

The five main considerations are, in order: (i) there is a relevant heterogeneity among environmental effects. Thus, they are time-varying and change significantly year by year; (ii) heterogeneity also matters across units, and thus these effects heterogeneously affect sharks’ prey preferences over time; (iii) non-linear relationships between predictors and outcomes need to be quantified through appropriate econometric algorithms; (iv) the environmental factors all have a strong effect on WSs’ prey preferences; (v) potential environmental problems can be addressed in a particular time. For instance, we are interested to investigate why WSs would prefer the tuna bait rather than the seal-shaped decoy in 2008 and 2011. The last two points are addressed in the second and third step, respectively.

### 3.3. LOGIT Regression Model

The second step, the LOGIT regression model (Table 3), highlights the main environmental factors causing the WSs to prefer the seal-shaped decoy instead of the tuna bait.

Table 3 displays the estimation outputs, where the significance level is α=5% (as a default). The five main findings are, in order: (i) all the results displayed in the first step are significant; (ii) all predictors, except ‘temperature’, positively affect the variable of interest ‘prey’. For instance, better sea conditions increase the seal-shaped decoy preferences across sightings. The same accounts for high tide and better underwater visibility. On the contrary, higher water temperature increases the tuna bait preferences across sightings; (iii) according to the magnitude of the model, the factor that most affects the outcome is ‘tide’ (2.10), followed by ‘visibility’ (0.83), ‘temperature’ (−0.66), and ‘sea’ (0.65). The constant has not been considered, displaying an effect on the outcome, with all predictors being fixed ($X_{i}=0$); (iv) ‘visibility’ and ‘oktas’ predictors are strongly negatively correlated (−43%) and show a significant linear relationship (Figure 2). Thus, by dropping ‘oktas’ within the system, any (potential) collinearity problems are dealt with, and we obtain better predicted outcomes; (v) the variable ‘year’ is also significant and positive, highlighting the presence of relevant endogeneity issues affecting sharks’ predatory behavior and heterogeneous effects given the unexpected environmental effects.

### 3.4. Sample Marginal Effects

The third and final step, the sample marginal effects (Table 4), evaluates major environmental factors affecting feeding behavior.

Table 4 displays the estimation outputs, where the significance level is α=5% (as default). Three relevant findings are addressed: (i) the main factors affecting the outcome correspond to the ones found in the logistic function (second step). More precisely, in order of importance, we have: (1) ‘tide’, (2) ‘visibility’, (3) ‘temperature’, and (4) ‘sea’; (ii) the marginal effect values appear sensible. For instance, a one-unit change associated with an observation increases the probability of preferring the seal-shaped decoy by 61% with high ‘tide’, 55% with better ‘visibility’, 51% with less water temperature (‘temp.’), and 49% with better ‘sea’ conditions; (iii) the factor ‘year’, though it affects the outcome with lower associated probability, is significant and must be included in the analysis when studying environmental problems.

Finally, we look more deeply into these sample marginal effects, referring to the average individual effects to investigate the same effects in every period (Table 5). We focus on 2008 and 2011, representing the time periods in which white sharks preferred the tuna bait rather than the seal-shaped decoy [10]. In 2011, according to the total sightings, tide events were exclusively characterized by low tide and the highest water temperature was registered. The same results were found in 2013, with two key differences: (i) in 2011, the total sightings were more than double (63 vs. 29), highlighting the highest sample marginal effects according to the predictor ‘tide’; (ii) the water temperature (the third most significant outcome-affecting factor) was, on average, balanced in 2013 compared to the other time periods, whereas in 2011 the temperature held at 18 °C. In 2008, the most sightings were characterized by low tide (25 on 40). Similar effects were observed in 2009, showing the same visibility and water temperature. However, in 2009, tide ranges were similar (12 observations occurred during low tide and 10 during high tide out of 22 total sightings), showing that, once again, ‘tide’ denotes the most significant/important factor in terms of affecting sharks’ predatory behavior.

## 4. Discussion

This study demonstrates that prey discrimination of bait-attracted WSs in Gansbaai is affected by environmental factors that drive the sharks to detect and choose prey depending on the conditions. Passive preys (tuna bait and seal-shaped decoy) do not show spatial autocorrelation, as demonstrated by the Moran’s Test, confirming that the odor from the tuna bait did not serve as chum within the 10 m range of the experiment. In Gansbaai, as stated by Micarelli et al. [10] from 2008–2013, the majority of WSs transient population, immatures included, were attracted by the seal-shaped decoy rather than by the tuna bait, except in the two years of 2008 and 2011. As observed by the first PERMANOVA test (Table 1), ‘visibility’ was not significant and ‘oktas’ displayed larger *p*-values and lower Pseudo F than other variables due to a multicollinearity problem, since underwater visibility is strongly correlated (−43%) with the light levels, as stated by the linear regression model in Figure 2. For that reason, light levels were omitted in the second PERMANOVA test (Table 2). As suggested by Strong [12], one of these aspects is a consequence of the other: low light levels (represented by high oktas values) reduce the surface underwater visibility that disadvantages the seal’s ability to detect the WS coming from below.

We observed a relevant heterogeneity among environmental effects that undergoes large changes year by year and affect sharks’ prey choice over the time. According to the sample marginal effects, the factors influencing the prey discrimination and the choice of the seal-shaped decoy for a one-unit change associated with the observation (Table 4) are, in decreasing order, tide ranges (61%), underwater visibility (55%), water temperature (51%), and sea conditions (49%).

Tide range is the most important environmental factor that influences prey choice. At Seal Island in South Africa, Hammerschlag et al. [2] stated that attack frequency to Cape fur seals increased significantly during high tides within 400 m of the island and over a depth range of 5–31 m where seals swim over shallow reefs. During high tide, the minimum depth from which sharks could approach seals was closer to shore, increasing the availability of prey and the possibility of predation. Additionally, at the Farallon Islands in California, WSs’ attacks increased with high tide that reduced the haul-out area for elephant seals, forcing them into the water and increasing predation events [11,16,27]. In fact, during high tide events, the number of Cape fur seals around our boat anchored at 100–150 m from Dyer Island increased, inducing sharks to use their visual acuity both to attack them and our seal-shaped decoy. For that reason, we propose that high tide could induce WSs from Gansbaai to turn their attention towards seal-shaped decoys rather than towards the tuna bait, consequently obtaining more caloric sources over low-energy ones, according to the theory of optimal foraging [28,29].

It is true that most of the years (2008, 2009, 2011, and 2013) were characterized by low tide, but it is also necessary to observe other environmental parameters that influence prey choice, such as underwater visibility and water temperature.

Underwater visibility is the second environmental factor that influences the sharks’ discrimination of prey and, on average, was good during all the years (at a visibility range registered from 1 to 5 m during all the years, minimum average visibility was 2.55 m in 2008 and 2009; maximum average visibility was 3.55 m in 2013). Higher average visibility and, consequently, low oktas values seemed to facilitate the attack on the seal-shaped decoy in Gansbaai. Our findings do not support the Seal Island results of Hammerschlag et al. [2] that observed a higher predatory success of WSs on Cape fur seals during low light levels and low underwater visibility. This inconsistency could be linked to two reasons: (i) stalking and ambush with low visibility are important for successful prey capture in many fishes, as well as sharks, including *C. carcharias* [12,30,31,32]. However, a depth range of 26–30 m around Seal Island is optimal for sharks to remain undetected and stalk seals from below with low visibility and enough vertical distance to build the perfect moment for attack and surface strike, as observed by Hammerschlag et al. [2]; on the other hand, around the Dyer Island Nature Reserve and in the Shark Alley channel that divides Dyer Island from Geyser Rock, the maximum depth is ca. 8–10 m and the vertical distance is not enough to surprise seals by exploiting the low visibility. For that reason, most of the attacks to passive preys at Gansbaai were horizontal [15] and in the presence of a good average underwater visibility. Sometimes, as observed by Huveeners et al. [17] in Neptune Island (South Australia), sharks hunting near the surface orient the sun directly behind them during approach, directly illuminating prey and increasing their detectability. (ii) As demonstrated by Micarelli et al. [10], vision plays an important role in mature and immature sharks with a dietary shift in their feeding pattern, and a good average visibility could help them to more consistently secure higher-energy prey sources, such as our shaped decoy.

Water temperature is the third environmental factor that influences prey selection and plays an important role in the physiology of ectotherms, regulating their internal functions, distribution, foraging, reproduction, prey distribution, and abundance [14]. Unlike most sharks and rays, the white shark is an endothermic animal within the family Lamnidae, and water temperature has an influence on its prey distribution rather than on the shark’s physiology [19]. Warmer water resulting in blooms of diatoms is associated with an increase in the number and diversity of teleosts, which are prey of white sharks [14]. Different tuna species, used as passive prey to attract white sharks, are present in South African waters and around Gansbaai, such as Albacore or Longfin tuna *Thunnus alalunga* (Bonnaterre, 1788), Yellowfin tuna *Thunnus albacares* (Bonnaterre, 1788), Big-Eye tuna *Thunnus obesus* (Lowe, 1839)*,* and Skipjack tuna *Katswonus pelamis* (Linnaeus, 1758) [33]. All of these species, except the Albacore, are mostly found off of the Southern Cape Coast of South Africa from spring to summer (www.deep-sea-fishing.co.za, accessed on 15 April 2022) when waters are warmer; their higher availability and abundance during these seasons and during our observation period in which the water was warmer likely influenced the tuna bait choice. Although WSs generally prefer to feed on juvenile Cape fur seals that leave Geyser Rock to forage offshore for the first time during the autumn and winter season, warmer water induces phytoplankton blooms that reduce the underwater visibility, hindering the shark’s ability to detect the odorless seal-shaped decoy and encouraging the tuna bait choice.

According to what has been said, warmer water temperatures registered in 2011 (18 °C on average) were associated with only low tide events (63), one of the lowest average underwater visibilities (2.79 m), and the highest number of interactions (63) that could have influenced the choice of the tuna bait. Additionally, in 2008 sharks preferred the tuna bait, even if the average water temperatures were the lowest registered (13.25 °C), but most of the interactions were characterized by low tide events (25 out of 40 total interactions) and by the lowest average underwater visibility (2.55 m), forcing the sharks to choose the tuna bait. During the other years (2009, 2010, 2012, and 2013) sharks preferred the seal-shaped decoy. In 2009, tide ranges were similar between high and low (10 vs. 12, respectively). The average water temperature in 2009 was also slightly higher than 2008 (14 °C vs. 13.25 °C, respectively) despite having the lowest average underwater visibility (2.55 m), justifying the choice, since both total sightings were the lowest registered during the entire study (22) and tide range remained the main influencing factor. In 2010, mostly high tide events were recorded (35 out of 54 total interactions) with a cool average water temperature (15 °C) and good average underwater visibility (2.83 m), justifying the seal-shaped decoy choice. In 2012, the situation was like 2009, since the number of low and high tide events were identical (16); the average water temperature was cool (15.2 °C), the underwater average visibility was good (3 m), and total sightings were few (32). In 2013, high tide events were not observed, the average water temperature was cool (15 °C) and total sightings were few (29), but average underwater visibility was the best out of all the years (3.55 m), probably justifying the seal-shaped decoy choice.

Ultimately, sea condition is the variable that affects prey selection the least. In any case, sea condition was, on average, slightly rough during all the years, except for 2013 and 2011 where it was calm. Wind intensity has a direct effect on the creation of rough conditions, and, therefore, could propel pinnipeds’ chemical stimuli (e.g., excreta, blood, organic compounds) from the Dyer Island Nature Reserve to the open sea, helping the sharks to locate the area where the seals are concentrated, as observed in the study of Strong [12] and Hammerschlag et al. [2] for Seal Island. At the same time, pinnipeds are forced to swim against the current with rough sea, producing sounds and losing their ability to maintain subsurface vigilance, which gives the sharks greater possibilities to strike [12]. However, it is also true that we used an odorless seal-shaped decoy to attract the sharks. A calm or slightly rough sea also has a direct effect on underwater visibility that, in our study, is the second most important environmental factor to influence prey discrimination, helping the sharks in Gansbaai to be able to visually detect the seal decoy more successfully. Notably, 2011 saw an average calm sea condition (1.94), but the other environmental factors, such as tide ranges events, underwater visibility, and water temperature, had a greater influence on the choice of the prey, which, in this case, was tuna. On the other hand, we registered the roughest average sea condition in 2008 (2.45 in terms of mean), alongside the lowest underwater visibility, the lowest water temperature, and a higher number of low tide events, justifying the tuna bait selection.

## 5. Conclusions

Environmental factors influence the WSs’ sensory ecology to choose between passive prey types during interactions. The choice of the seal-shaped decoy is strictly related to high tide events, good underwater visibility, cooler water temperature, and calm or slightly rough sea. The last three conditions highlight the importance of the mature and immature WSs’ visual ability in detecting the prey. WSs are initially attracted by the olfactory trace emitted by the tuna bait, but they subsequently shift their focus to the more calorically rich seal prey, helped by good environmental conditions. High tide, on the other hand, gives the sharks more chances to get closer to the island and reduces the prey haul-out area, increasing the possibility of predation on seals and the seal decoys alike. Some environmental factors were not taken into consideration during the study, such as wind direction, currents, and distance from the island, and could, therefore, be considered as topics of future inquiry at this site.

## Figures and Tables

**Figure 1 animals-12-03276-f001:**
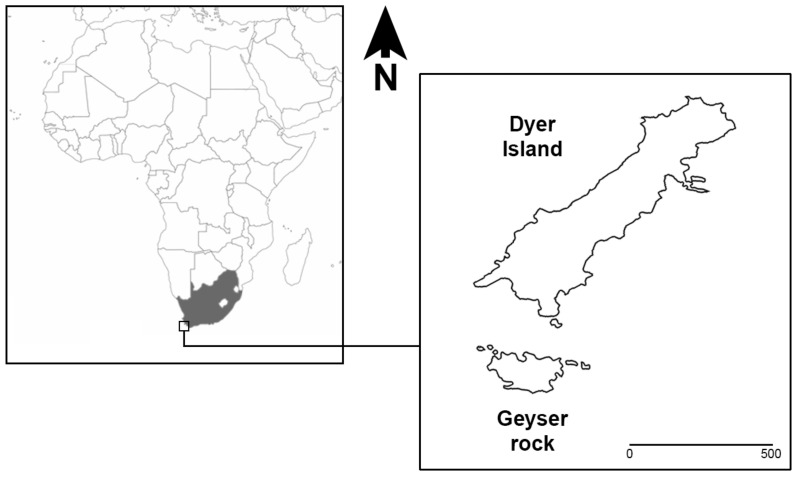
Dyer Island and Geyser Rock Nature Reserve.

**Figure 2 animals-12-03276-f002:**
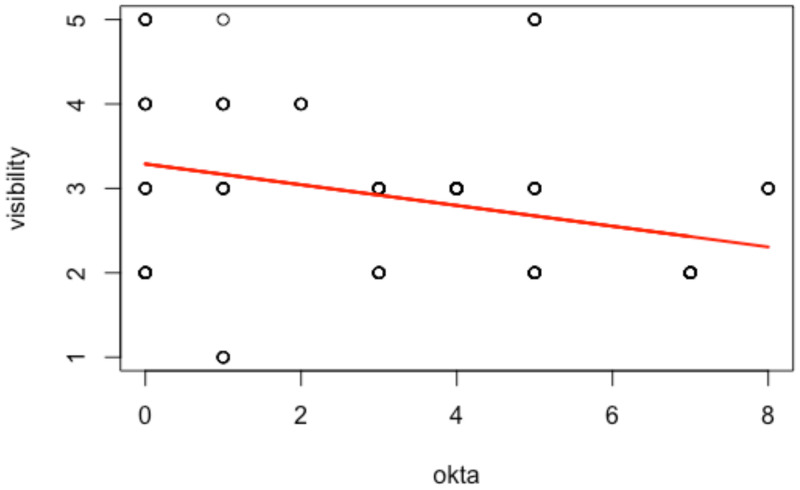
Scatter plot by regressing ‘okta’ (independent variable) on ‘visibility’ (dependent variable). The estimation output is negative and significant, displaying a consistent relationship between the variables. The scatter plot shows a large component of outliers affecting the output due to omitted variables and unobserved heterogeneity (endogeneity issues) dealt with the logistic function. The significance level used in the analysis is α=5% (as default).

**Table 1 animals-12-03276-t001:** PERMANOVA analysis between the elements ‘years’ and ‘effects’ across units. Here, the labels stand for ‘degrees of freedom’ (df), ‘sum of squared dissimilarities’ (SS), ‘Pseudo F statistic’ (Pseudo-F), and ‘associated *p*-value’ (Pr (>F)). The significance levels are: (**) significance at 5%; and (***) significance at 1%.

Source	df	SS	Pseudo-F	Pr (>F)
Prey	1	9.8800 × 10^−7^	73.2246	0.001 ***
Oktas	1	4.2400 × 10^−7^	4.8126	0.031 **
Sea	1	3.6560 × 10^−6^	61.5374	0.001 ***
Tide	1	2.8770 × 10^−6^	82.6926	0.001 ***
Visibility	1	4.0000 × 10^−8^	45.3016	0.500
Temp.	1	7.4930 × 10^−6^	77.1439	0.001 ***
Residual	233	1.5005 × 10^−7^		
Total	239	3.6743 × 10^−7^	1	

**Table 2 animals-12-03276-t002:** PERMANOVA analysis between the elements ‘years’ and ‘effects’ across units, dealing with multicollinearity problems. Here, the labels stand for ‘degrees of freedom’ (df), ‘sum of squared dissimilarities’ (SS), ‘Pseudo F statistic’ (Pseudo-F), and ‘associated *p*-value’ (Pr (>F)). The significance level is: (***) significance at 1%.

Source	df	SS	Pseudo-F	Pr (>F)
Prey	1	7.4628 × 10^−8^	84.1295	0.001 ***
Sea	1	4.3409 × 10^−7^	65.3846	0.001 ***
Tide	1	4.1240 × 10^−7^	87.9824	0.000 ***
Visibility	1	5.0832 × 10^−8^	73.3625	0.000 ***
Temperature	1	8.1254 × 10^−7^	81.9472	0.000 ***
Residual	233	3.8756 × 10^−7^		
Total	238	4.9857 × 10^−7^	1	

**Table 3 animals-12-03276-t003:** Logit regression model accounting for all the five environmental factors across units over time. Here, ‘Coefficients’ refers to the factors within the model; ‘Estimate’ denotes $b_{k}$ (the estimated $\beta_{k}$); ‘SE’ stands for standard error; ‘z-value’ denotes the test statistic obtained for each predictor; and ‘Pr (>|z|)’ refers to the associated *p*-value in a two-sided hypothesis test (where the null accounts for non-significance). The significance levels are: (**) significance at 5%; and (***) significance at 1%.

Coefficients	Estimate	SE	z-Value	Pr (>|z|)
Costant	−1049.7508	269.6977	−3.892	9.93 × 10^−5^ ***
Sea	0.6342	0.3426	2.040	0.041335 **
Tide	2.0951	0.3536	5.925	3.12 × 10^−9^ ***
Visibility	0.8361	0.2743	3.048	0.002303 ***
Temperature	−0.6630	0.3250	−1.910	0.041689 **
Year	0.5203	0.1343	3.874	0.000107 ***

**Table 4 animals-12-03276-t004:** Sample marginal effects for each observation unit, given n observations, are accounted for. Here, ‘Coefficients’ refers to the factors within the model; ‘dF/dx’ denotes the partial derivatives displaying the marginal effects of the predictors ($x_{ik}^{\prime }$) on $y_{i}$ (‘prey’); ‘SE’ stands for standard error; ‘z-value’ denotes the test statistic obtained for each predictor; and ‘Pr (>|z|)’ refers to the associated *p*-value in a two-sided hypothesis test (where the null accounts for non-significance). The significance level is: (***) significance at 1%.

Coefficients	dF/dx	SE	z-Value	Pr (>|z)
Sea	0.492973	0.034583	3.5558	0.0003768 ***
Tide	0.610744	0.058782	6.9875	2.798 × 10^−12^ ***
Visibility	0.557756	0.035013	4.5056	6.618 × 10^−6^ ***
Temperature	−0.512029	0.037561	−2.9826	0.0028581 ***
Year	0.157797	0.020775	4.2261	2.377 × 10^−5^ ***

**Table 5 animals-12-03276-t005:** Summary and descriptive table for the average individual (total sightings) over time. Here, ‘Tot. Sigh.’ stands for total sightings, and ‘W. Temp.’, ‘U. Visibility’, and ‘Sea Cond.’ refer, in average ± SD, to water temperature (°C), underwater visibility (m), and sea conditions, respectively. The ranges would correspond to the minimum and maximum values displayed in the table.

Years	Tot Sigh.	Low Tide	High Tide	W. Temp. (Mean ± SD)	U. Visibility (Mean ± SD)	Sea Cond. (Mean ± SD)
2008	40	25	15	13.25 ± 1.83	2.55 ± 0.33	2.45 ± 0.35
2009	22	12	10	14 ± 1.08	2.55 ± 0.33	2.45 ± 0.035
2010	54	19	35	15 ± 0.08	2.83 ± 0.05	2.17 ± 0.07
2011	63	63	0	18 ± 2.93	2.79 ± 0.09	1.94 ± 0.16
2012	32	16	16	15.2 ± 0.12	3 ± 0.12	2 ± 0.1
2013	29	29	0	15 ± 0.08	3.55 ± 0.67	1.59 ± 0.51

## Data Availability

https://www.researchgate.net/project/Great-White-Shark-Carcharodon-carcharias-Behaviour-Ecology-and-cotoxicology/update/61312b952897145fbd6df39c accessed on 10 April 2022.

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
