# Peer review of "Influence of Environmental Factors on Prey Discrimination of Bait-Attracted White Sharks from Gansbaai, South Africa"

_animals, 2022, doi:10.3390/ani12233276_

Round 1
Reviewer 1 Report
Authors aim to contribute with further analyses of a data set of feeding interactions of white sharks previously worked by Micarelli et al. (2021). In this sense, some methodological aspects must be considered, including the experimental design and the choice of statistical tests. The major problem is in sampling design. Authors failed to demonstrate in the Materials and Methods that the stimuli of the decoy and the tuna bait were isolated from each other at the 10m range of the experiment. This is an important premise for testing the hypotheses because the odor from the tuna bait possibly could also serve as chum within this range. Authors should use a statistical test (cf. Moran's test) to demonstrate that the samples do not show spatial autocorrelation, or else, provide such statement from previous studies, since they mention that this research protocol had already been employed (cf. Sperone et al. 2012, Micarelli et al. 2021). If such test or statement fails to confirm the independence of samples, authors should point out the limitations of their sampling design in the Discussion.
Regarding the statistical procedures, it is not clear why the authors have converted water temperature into a categorical variable, and why they have used two permanovas for assessing multicollinearity and filtering variables for the Logit model, instead of using a Variation Inflation Factor (VIF) as an “a priori” test.
In the Results section, units of measurements and standard deviation of sample measn should be givem [cf. Table 5, lines 387-390]. Other issues are pointed directly as comments in the manuscript file.

Reviewer 2 Report
Review of Animals-2027254 “Influence of Environmental Factors on Prey Discrimination of Bait-Attracted White Sharks from Gansbaai, South Africa”
Summary:
Romana Reinero et al. assess the influence of environmental factors on prey discrimination of bait-attracted white sharks over a six-year period (2008-2013) at Dyer Island Nature Reserve (Gansbaai, South Africa). Romana Reinero et al.’s results demonstrated both immature and mature white sharks were attracted by the seal shaped decoy rather than the tuna bait, and tide ranges, underwater visibility, water temperature, and sea conditions were the factors which drove white sharks to select the seal shaped decoy. Finally, this study confirms the importance of visual ability in mature and immature white sharks with 22 dietary shifts in their feeding pattern.
General comment:
Strengths: Study the influence of environmental factors on predator-prey interactions, as very few studies have previously analysed the interaction but have never specifically focused on bait attraction.
Weaknesses: presentation of the results
The hypotheses, methodology, results, discussion and conclusion are generally well presented, except for some sections that need to be improved. The manuscript needs some minor changes before being accepted. Details are listed below:
Abstract:
Line 25: Please add an introduction line on the topic: see e.g. Lines 17-21
Introduction
Line 65: please, indicate the country of Seal Island, as done for Neptune Island in Line 63
Materials and Methods
Lines 125: it is not clear here if the materials &methods are ex-novo information and data were collected by the authors of the manuscript or retrieved from the previous work of Micarelli et al, as mentioned in Line 89. If the latter, please clearly mention it and cite the work.
Lines 144-145: it is not very clear. one specimen or a specimen? Please check and rephrase accordingly
Results
Line 275: The paragraph cannot start either with a table or “Table 1 displays…” Please, add a general sentence introducing the results.
Line 286: please add the full name for SS and Pseudo-F
Line 312: The paragraph cannot start either with a table or “Table 3 displays…” Please, add a general sentence introducing the results.
Lines 322 and 327: please replace “temp.” with “temperature. Same in the Table 3
Line 344: The paragraph cannot start either with a table or “Table 4 displays…” Please, add a general sentence introducing the results.
Line 361: please replace “temp.” with “temperature. Same in the Table 4
